# *Bacillus*-Based Probiotic Treatment Modified Bacteriobiome Diversity in Duck Feces

Natalia B. Naumova [1,*], Tatiana Y. Alikina [1], Natalia S. Zolotova [2], Alexey V. Konev [2], Valentina I. Pleshakova [2], Nadezhda A. Lescheva [2] and Marsel R. Kabilov [1]

[1] Institute of Chemical Biology and Fundamental Medicine, Siberian Branch of the Russian Academy of Sciences, Lavrentieva 8, 630090 Novosibirsk, Russia; alikina@niboch.nsc.ru (T.Y.A.); kabilov@niboch.nsc.ru (M.R.K.)

[2] The Faculty of Veterinary Medicine, Stolypin Omsk State Agrarian University, Institutskaya pl. 1, 644008 Omsk, Russia; ns.zolotova@omgau.org (N.S.Z.); av.konev@omgau.org (A.V.K.); vi.pleshakova@omgau.org (V.I.P.); lescheva@list.ru (N.A.L.)

\* Correspondence: naumova@niboch.nsc.ru

**Abstract:** The intestinal health of poultry is of great importance for birds' growth and development; probiotics-driven shifts in gut microbiome can exert considerable indirect effect on birds' welfare and production performance. The information about gut microbiota of ducks is scarce; by using high throughput metagenomic sequencing with Illumina Miseq we examined fecal bacterial diversity of Peking ducks grown on conventional and *Bacillus*-probiotic-enriched feed. The probiotic supplementation drastically decreased the presence of the opportunistic pathogen *Escherichia/Shigella*, which was the major and sole common dominant in all samples. Seventy other bacterial species in the ducks' fecal assemblages were found to have probiotic-related differences, which were interpreted as beneficial for ducks' health as was confirmed by the increased production performance of the probiotic-fed ducks. Bacterial α-biodiversity indices increased in the probiotic-fed group. The presented inventory of the duck fecal bacteriobiome can be very useful for the global meta-analysis of similar data in order to gain a better insight into bacterial functioning and interactions with other gut microbiota to improve poultry health, welfare and production performance.

**Keywords:** 16S rRNA gene; amplicon sequencing; ducks; probiotic; gut microbiome





## 1. Introduction

Over 21 million ducks are raised for human consumption each year in the Russian Federation [1], yet so far no research has been conducted on the gut microbiome of the Pekin duck breed, maintained in the country. The intestinal health of poultry is of great importance for birds' welfare and hence their production performance, food safety and environmental consequences [2]. Industrial poultry production still relies on antibiotics as growth promoters, although probiotics nowadays are becoming an increasingly indispensable pharmacological component for production of high quality food [3]. Probiotic preparations can be based on different microorganisms, including the spore-producing ones like *Bacillus* [4,5], which are Gram-positive, aerobic, spore-forming bacteria ubiquitous in the environment. Importantly, they have high stability under adverse environmental conditions, which is indispensable for probiotic cells to survive processing and storage of feed, its passage through the gastrointestinal tract and subsequent chemical digestion processes. The antagonistic effect of such probiotics on the pathogenic gut microflora of humans and animals has been known since long ago [6]. However, most research about the effect of dietary administration of probiotic *Bacillus* strains on growth performance has been conducted in chicken, mouse, and pig [7–10], and yet only recently it was experimentally shown that certain strains of *B. subtilis* can provide beneficial effects on the growth of young broiler chickens and have the potential to replace antibiotic growth promoters [11] or

improve egg quality [12]. Similar studies on ducks have been fewer [13,14], and we failed to find reports on gut microbiome research using probiotic *Bacillus* strains for Pekin ducks.

Knowledge of the microbiome profiles in regional agricultural populations could help in drawing a global picture of duck gut microbiota, leading to a better insight into the regional effects of production technologies such as the use of probiotics, prebiotics, synbiotics, enzymes and antibiotics. As there is still a gap in knowledge concerning the effectiveness of probiotic supplementation in shaping gastrointestinal taxonomic profiles in ducks, the objective of the study was to examine composition and structure of ducks' gut bacterial assemblages by estimating diversity of phylogenetically significant fragments of 16S rRNA genes from the feces of ducks grown on conventional and probiotic-enriched feed by using high throughput metagenomic sequencing.

## 2. Materials and Methods

### 2.1. Duck Breed and Experimental Design

All experimental procedures involving ducks met the guidelines approved by the institutional animal care and use committee and were performed in accordance with the Russian National Law concerning the care of animals for research purposes, as well as in compliance with the European Commission Directive 2010/63/EU on the protection of animals used for scientific purposes [15]. Ducks *Anas platyrhynchos* of the Peking breed Agidel variety were raised and grown at a poultry farm in the Omsk region, Russia, and female ducks were used in the study. Until 10 days of age the ducklings were kept in stainless steel cages at +28–30 °C and 65–70% relative humidity on a small-mesh-flooring; after that they were put into a bigger house where they could freely roam on a deep pine-shaving-based litter at +25 °C. Then, the 30-d-old birds were placed into the premises with similar deep litter, air temperature of +14–20 °C, drinking water ad libitum and access to artificial ponds. From the first day of life to three weeks of age, the ducks were fed ad libitum with a starter diet (wheat, soya beans, oil free sunflower seed, sunflower seed cake, fish flour, methionine, threonine, lysine, sodium chloride, premix), providing 3100 kcal/kg of feed and 23% of crude protein. Then, from four to five weeks of age, the birds were fed with a grower diet with threonine substituted with cysteine and providing 3150 kcal/kg of feed and 21% of crude protein. Additionally, from 6 weeks of age until the end of the performance the ducks were fed with a similar diet but providing 3200 kcal/kg of feed and 20% of crude protein.

The ducks were assembled in two groups of sixteen birds in each. One group received conventional feed as described above supplemented with a probiotic (probiotic-fed) during the entire growth period of 60 d as per manufacturer's instructions, i.e., 0.4 kg/t during the first 15 d followed by 1.0 kg/t till the end of the growth. The other group received only conventional feed (control).

The commercially distributed probiotic preparation Olin®, produced for Probiotic-Plus LLC (Russia) [16], was used in the study. According to the manufacturer, the preparation contains dried biomass of antagonistically active strains of *Bacillus subtilis* and *Bacillus licheniformis*, registered in the Russian Collection of Industrial Microorganisms under accession numbers 10172 and 10135, respectively, with plate counts of at least $2 \times 10^9$ CFU per 1 g of the preparation [17].

### 2.2. Sample Collection

At sixty days of age, all birds were weighed, and five apparently healthy ducks were selected at random from each group, caught, not fed for 8 h, but could drink ad libitum, and then euthanized by cervical dislocation in compliance with the European Commission Directive 2010/63/EU on the protection of animals used for scientific purposes [15]. Within two hours the recta were opened using sterile scissors, and the contents were collected into sterile vials and frozen at −196 °C. In the laboratory the samples were stored at −80 °C prior to the DNA extraction.

### 2.3. Extraction of Total Nucleic Acid from Feces

Total DNA was extracted from 250 mg of feces using the DNeasy Powersoil Kit (Qiagen, Germany) as per the manufacturer's instructions [18] to lyse microbial cells and obtain high-quality DNA solutions free from PCR inhibitors. The bead-beating was performed using a TissueLyser II (Qiagen, Germany), for 10 min at 30 Hz. No further purification of the DNA was needed. The quality of the DNA was assessed using agarose gel electrophoresis.

### 2.4. 16S rRNA Gene Amplification and Sequencing

The 16S DNA region was amplified with the primer pair F343 (5′-TACGGRAGGCAG CAG-3′) and R803 (5′-CTACCAGGGTATCTAATCC-3′) combined with Illumina adapter sequences [19]. PCR amplification was performed as described earlier [20]. A total of 200 ng PCR product from each sample was pooled together and purified through a MinElute Gel Extraction Kit (Qiagen, Germany). The obtained libraries were sequenced with $2 \times 300$ bp paired-ends reagents on MiSeq (Illumina, San Diego, CA, USA) in the SB RAS Genomics Core Facility (ICBFM SB RAS, Novosibirsk, Russia). The read data reported in this study were submitted to the GenBank under the study accession PRJNA523560.

### 2.5. Bioinformatic and Statistical Analyses

Raw sequences were analyzed with the UPARSE pipeline [21] using Usearch v11.0. The UPARSE pipeline included merging of paired reads; read quality filtering; length trimming; merging of identical reads (dereplication); discarding singleton reads; removing chimeras and operational taxonomic unit (OTU) clustering using the UPARSE-OTU algorithm. The OTU sequences were assigned a taxonomy using the SINTAX [22] and 16S RDP training set v.16 [23].

Taxonomic structure of thus obtained sequence assemblages, i.e., a collection of different species at one site at one time [24], was estimated by the ratio of the number of taxon-specific sequence reads to the total number of sequence reads, i.e., by the relative abundance of taxa, expressed as a percentage.

Statistical analyses of the data were perfumed using Statistica v.13.3 software (Statsoft, Tulsa, OK, USA). Comparison of relative abundances of different bacterial taxa in fecal samples of the control and probiotic-fed group was carried out using the Mann–Whitney nonparametric test, whereas comparison of ducks' production characteristics ANOVA and Fisher's least significant difference test were carried out. The rarefaction curves were obtained using iNEXT 2.0.15 in R-package [25] and biodiversity indices calculated with the help of PAST 2.17 software [26].

## 3. Results

### 3.1. Taxonomic Richness and Structure of Duck Fecal Bacterial Assemblages

After 16S gene amplicon sequencing, quality filtering and chimera removal a total of 666,588 high-quality DNA sequences were obtained from feces of the 10 ducks. High-quality reads were clustered using >97% sequence identity into 568 bacterial operational taxonomic units (OTUs). The obtained sets of sequences for each sample were analyzed by plotting the number of OTUs against the total number of sequence reads (Figure 1). The resulting rarefaction curves demonstrated sufficient out coverage to describe the bacterial composition and compare assemblages of different groups [27].

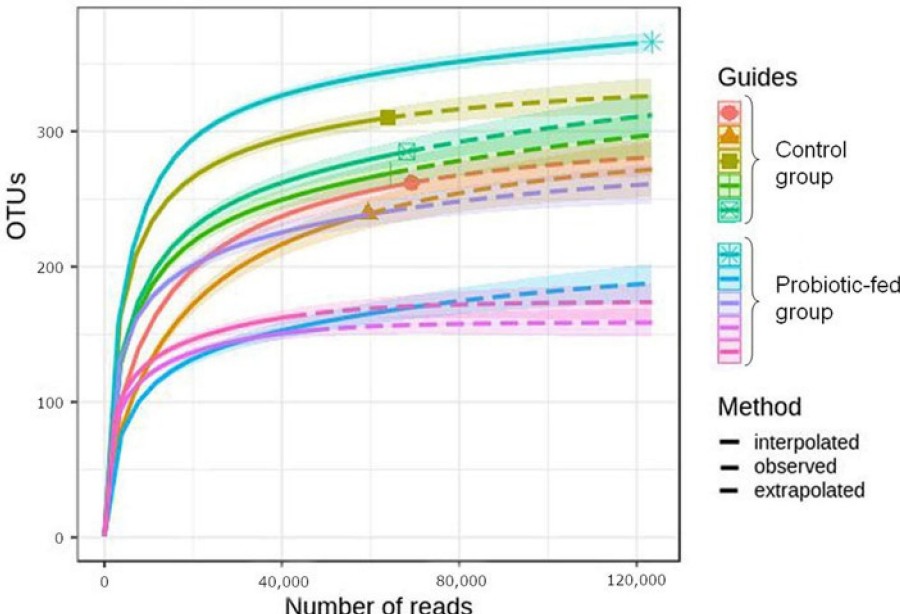

**Figure 1.** Rarefaction curves for the OTU number in fecal bacterial assemblages of the ducks.

The total number of different-level taxa identified in the study is shown in Table 1. The mooutOTU-rich phyla were *Proteobacteria* (147 OTUs, or 26% of the total number of identified OTUs) and *Firmicutes* with (134 OTUs, or 24%), followed by *Actinobacteria* (131 OTUs, or 23%) and *Bacteroideteout*25 OTU, or 4%). Taxonomic richness in the studied samples was found to drastically decrease if only the dominant members, i.e., the ones contributing at least 1% into the total number of sequence reads, of the bacterial assemblages, were taken into account (Table 1).

**Table 1.** Taxonomic richness of fecal bacterial assemblages of ducks.

| Taxon Level | All OTUs | Taxonomic Attribution | | |
|---|---|---|---|---|
| | | Dominant [a] OTUs | | |
| | | Both Groups | Control Group | Probiotic-Fed Group |
| Phylum | 15 | 4 | 3 | 3 |
| Class | 36 | 6 | 4 | 6 |
| Order | 63 | 6 | 4 | 6 |
| Family | 137 | 9 | 4 | 9 |
| Genus | 251 | 12 | 5 | 12 |
| OTU | 567 | 13 | 5 | 13 |

[a] OTUs were considered dominant if their relative abundance was more than 1%.

Two bacterial phyla—*Firmicutes* and *Proteobacteria*—collectively accounted for more than 90% of the total sequence reads in fecal assemblages (Figure 2a). The overwhelming majority of sequences represented three classes (Figure 2b), three orders (Figure 2c), just six families (Figure 2d) and six genera (*Escherichia/Shigella*, *Terrisporobacter*, *Streptococcus*, *Enterococcus*, *Romboutsia* and an unclassified representative of *Clostridiacea*e, Figure 3). The commercial probiotic preparation, fed to the ducks in the study, was found to contain 54 OTUs, with one OTU (*Bacillus* sp.) accounting for 58% of the total number of sequence reads, detected in the preparation. Other dominant components of the probiotic preparation were *Pseudomonas* spp. (three OTUs), *Comamonas* sp. (one OTU) and unclassified *Enterobacteriaceae* (two OTUs). These bacteria were practically absent in fecal bacteriobiomes of both groups. Overall *Bacillus* class was represented by seven OTUs in fecal assemblages of ducks (Figure 2b), collectively accounting for a tiny portion of the total

number of sequence reads (0.015% and 0.002% in the control and probiotic-fed groups, respectively).

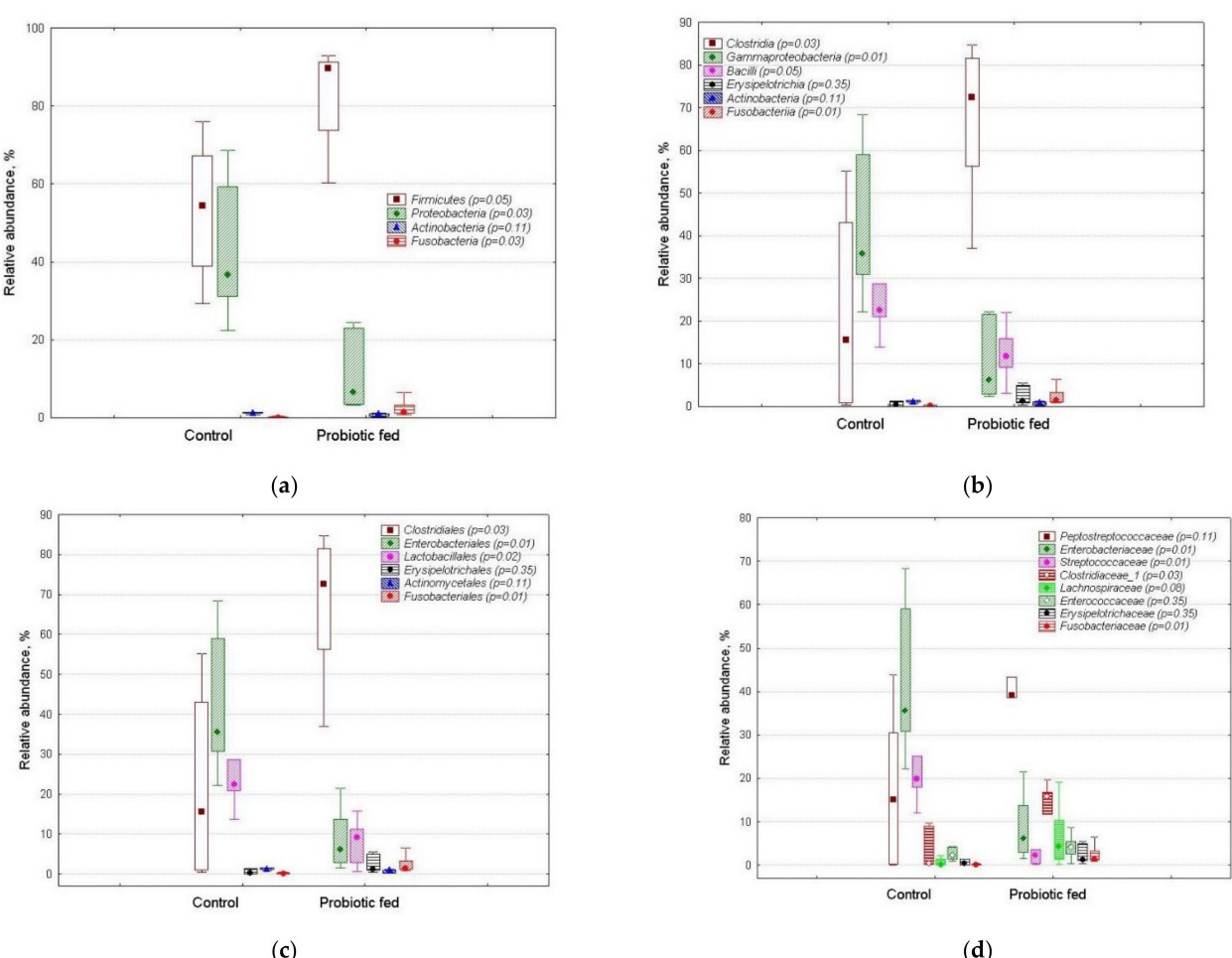

**Figure 2.** Relative abundance of taxon-specific sequences in fecal bacterial assemblages of ducks of the control and probiotic-fed groups: (**a**) phylum, (**b**) class, (**c**) order and (**d**) family levels. The markers show median, boxes show 25–75% percentiles, while the lines indicate fluctuation ranges. The *p*-values as estimated for each taxon by Mann–Whitney test are shown in brackets.

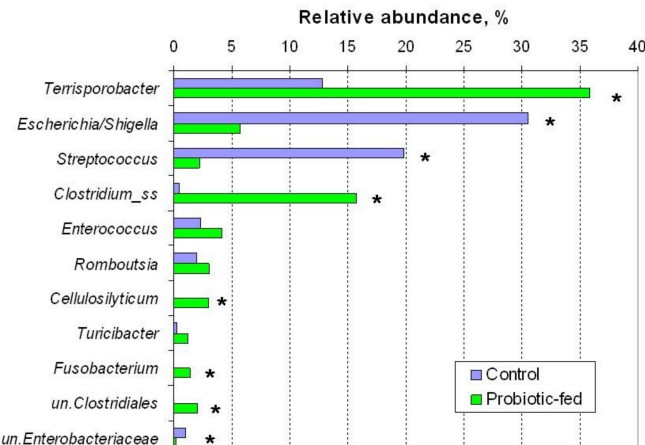

**Figure 3.** Relative abundance of genera in fecal bacterial assemblages of ducks of the control and probiotic-fed groups. Symbol * at the right of the columns denotes statistically significant difference between the groups (Mann–Whitney test, *p* ≤ 0.05). "un." stands for unclassified.

Of the total OTU number detected in the studied samples, only 13 OTUs, or 2%, were dominants, i.e., contributed $\geq 1\%$ to the total number of sequences (Table 1). The number of dominant OTUs per sample varied from three to nine in the control group, with just two OTUs being common for all samples (*Escherichia/Shigella* sp. and *Streptococcus* sp.). In the probiotic-fed group the number of dominant OTUs varied from 5 to 17 per sample, with three OTUs being common for all samples (*Escherichia/Shigella* sp., *Terrisporobacter* sp. and *Romboutsia sedimentorum*). Thus only one OTU, namely *Escherichia/Shigella* sp., was common for all studied samples. Its relative abundance varied from 30 to 68% in samples of the control group, and from 6 to 20% in samples of the probiotic-fed group.

### 3.2. OTUs' Relative Abundance in Duck Fecal Bacterial Assemblages

The relative abundance of some OTUs found in the bacterial assemblages of ducks is shown in Table 2. As mentioned above, the ultimate dominant in the control group was *Escherichia/Shigella* sp. The second major dominant OTU in the control group was *Streptococcus* sp. In fecal assemblages of the probiotic-fed group the abundance of this bacterium was almost 10 times lower.

**Table 2.** Relative abundance (%) of bacterial OTUs, dominant in the fecal assemblages of ducks of the control and/or probiotic-fed groups.

| | OTU | Control Group | Probiotic-Fed Group | *p*-Value |
|---|---|---|---|---|
| 1 | *Escherichia/ Shigella sp.* [1] | 30.6 | 5.7 | 0.012 |
| 2 | *Terrisporobacter sp.* | 12.9 | 36.4 | 0.037 |
| 3 | *Streptococcus sp.* | 20.0 | 2.2 | 0.012 |
| 6 | *Enterococcus cecorum* | 1.2 | 3.8 | 0.210 |
| 7 | *unc. Clostridiaceae_1* [2] | 0.2 | 3.7 | 0.012 |
| 8 | *Clostridium_ss[3]sp.* | 0.3 | 1.5 | 0.295 |
| 10 | *Cellulosilyticum sp.* | 0.1 | 1.3 | 0.094 |
| 11 | *Turicibacter sanguinis* | 0.3 | 1.2 | 0.403 |
| 12 | *unc. Clostridiales* | 0.01 | 1.9 | 0.094 |
| 13 | *Clostridium_ss[3]sp.* | 0.04 | 1.04 | 0.210 |
| 14 | *Fusobacterium sp.* | 0.11 | 1.41 | 0.012 |
| 15 | *Cellulosilyticum lentocellum* | 0.04 | 1.48 | 0.037 |
| 19 | *Romboutsia sedimentorum* | 2.0 | 3.2 | 0.403 |

[1] The lines with statistically significant ($p \leq 0.05$) difference are highlighted in bold; [2] "unc." stands for unclassified; [3] "ss" stands for *sensu stricto*.

Overall, 70 OTUs were found to have differential relative abundance ($p \leq 0.05$) in fecal microbiota of the studied groups. Most of these OTUs were minor or rare members, contributing much less than 1% into the total number of sequence reads.

Four dominant OTU were found to have increased ($p \leq 0.05$) abundance in fecal bacteriobiomes of the probiotic-fed group. *Terrisporobacter* sp. sequences were almost three times more abundant in the probiotic-fed group, comprising one third of the entire bacteriobiome. A *Fusobacterium* sp. was also found increased in the probiotic-fed group (Table 2).

### 3.3. Biodiversity Indices of the Duck Fecal Bacterial Assemblages

Biodiversity indices serve to compact information about communities, assemblages, guilds, etc. of living organisms; thus, the indices are useful for comparing large arrays of metagenomic data. Therefore, for each studied sample, i.e., an array with the number of sequence reads for each OTU, we calculated $\alpha$-biodiversity indices (Table 3).

**Table 3.** Alpha-biodiversity indices (median) of fecal bacterial assemblages of ducks of the control and probiotic-fed group.

| Index | Control Group | Probiotic-Fed Group | *p*-Value |
|---|---|---|---|
| Total number of identified OTUs | 208 | 114 | 0.095 |
| Dominance (D) | 0.31 | 0.17 | 0.222 |
| Simpson (1-D) | 0.69 | 0.83 | 0.222 |
| Shannon | 1.92 | 2.49 | 0.151 |
| Evenness [1] | 0.02 | 0.06 | **0.032** |
| Brillouin | 1.91 | 2.49 | 0.151 |
| Menhinick [1] | 0.85 | 0.51 | **0.032** |
| Margalef | 19 | 9 | 0.095 |
| Equitability | 0.33 | 0.49 | 0.095 |
| Fisher-alpha | 27 | 13 | 0.095 |
| Berger-Parker | 0.46 | 0.36 | 0.841 |
| Chao-1 | 246 | 127 | 0.095 |

[1] The lines with statistically significant ($p \leq 0.05$) difference are highlighted in bold.

The probiotic-fed group showed a tendency for decreased OTU richness, as indicated by the number of OTUs, Chao-1, Fisher's alpha, Margalef and Menchinik indices, and increased evenness, with Shannon and Brilluin indices, on the contrary, tending to increase in the probiotic-fed group.

### 3.4. Production Performance of Ducks

The data on production characteristics for the entire groups, i.e., consisting of 16 birds each, were normally distributed: ANOVA showed that probiotic supplementation accounted for 43% of the bird body mass variance at day 60 and for 30% of the growth rate variance. Thus due to the beneficial effect of probiotic-enriched feed ducks' production characteristics improved, as the probiotic-fed ducks demonstrated (Table 4) higher both daily mass increase rate (by 4.0 g/bird) and total body mass at the end of the feeding (by 235 g/bird).

**Table 4.** Production characteristics of ducks fed with conventional (control) and probiotic-supplemented feed (probiotic-fed group).

| Characteristic | Control Group (*n* = 16) [2] | Probiotic-Fed Group (*n* = 16) | *p*-Value [1] |
|---|---|---|---|
| Living mass of a 1-day-old duck, g/bird | 57.8 ± 5.8 [2] | 57.9 ± 6.3 | 0.931 |
| Living mass of a 60-day-old duck, g/bird | 2772 ± 222 | 3007 ± 141 | **0.001** |
| Average daily gain, g/bird per day | 45.2 ± 3.7 | 49.2 ± 2.3 | **0.001** |
| Feed intake, kg/kg bird mass | 3.35 | 2.85 | |

[1] The lines with statistically significant ($p \leq 0.05$) difference are highlighted in bold. [2] Performance was assessed for a bigger sets of birds than the fecal microbiome.

Supplementing conventional duck feed with probiotic resulted in 0.5 kg less consumption per 1 kg of duck living mass. The ducks in the control group consumed on average 0.72 kg more feed, as compared to the probiotic-fed ducks.

### 4. Discussion

The finding that two bacterial phyla, namely *Firmicutes* and *Proteobacteria*, prevailed in the ducks' feces agrees with the results obtained in other studies: for instance, the representatives of *Firmicutes*, *Proteobacteria* and *Bacteroidetes* were reported to account for at least 90% of bacteriobiomes in duck's ileum and cecum [28–31]. As for the rectum (as in our study), recently *Firmicutes* and *Proteobacteria* were found to account for 75% of the Muscovy ducks' rectum bacteriobiome [32], with *Proteobacteria* abundance being twice lower as in the control group in our study (15% vs. 30%). The difference may be attributed

to the difference in duck species and sex, as well as other factors; this is an area that is still poorly investigated.

The fact that we did not explicitly detect *B. subtilis* and *B. licheniforms*, the major components in the probiotic preparation, may be due to the relatively short V3–V4 amplicon sequences, not allowing discrimination between closely related species [33]. Our result that the probiotic bacteria consumed by the ducks with their feed did not reside in the gut complies with the fact that *Bacillus* representatives are not common for the gut microbiota of poultry [31,34].

The occurrence of an *Escherichia/Shigella* bacterium in the probiotic-fed group was more than five times lower. The genus represents important pathogens of humans and animals [35], therefore the change was most likely beneficial for birds' health and welfare. Among the *Streptococcus* genus some serious pathogens for humans and animals were reported before: for example, the ones capable of causing meningitis in ducks [36]. Thus it seems that *Streptococcus* sp. in our study was not a beneficial bacterium, so decrease in the probiotic-fed ducks might have contributed to their enhanced production performance. The drastically decreased abundance of these two harmful bacteria, i.e., *Escherichia/Shigella* sp. and *Streptococcus* sp., in the probiotic-fed group confirms the antagonistic and hence beneficial impact of the *Bacillus*-based probiotic on the major opportunistic pathogens of the fecal microbiota of ducks. It should be noted that the ducks in this study harboring abundant *Escherichia/Shigella* and *Streptococcus* were apparently healthy, which means that even high abundance of a potential pathogen's sequence reads in a bacteriobiome is not be immediately manifested as an actual instance of a disease.

Although some authors claim that *Riemerella anatipestifer* is one of the most common bacterial pathogens of ducks [32], in our study none of the assigned OTUs were classified into *Riemerella*.

The finding that most of the differentially abundant OTUs in fecal bacterial assemblages of the conventionally and probiotic-fed groups were minor or rare members suggests that low-abundant OTUs may be important for the host adjustments to shifts in environmental conditions. Such OTUs in the gut microbiome may have systemic interactions with potentially important consequences for the microbial performance within a host organism.

There is evidence about the pathogenicity of the *Terrisporobacter* genus for humans [37], but for animals and poultry we could not find such information. Some *Terrisporobacter* genus representatives are known as chemoorganotrophs, while others are chemolithoautotrophs, or acetogenic bacteria [38,39], capable of decomposing plant material in anaerobic conditions. Thus this bacterium is beneficial for host functioning, and its increased abundance in the probiotic-fed ducks' feces also confirms the positive effect of the *Bacillus*-based probiotic on the fecal microbiota of ducks.

Although recently a *Fusobacterium* sp. was reported to be associated with decreased production of hens, thus likely being an opportunistic pathogen [40], *Fusobacteria* phylum representatives are common and often dominant members of the gut microbiota of wild ducks and geese [41]. In view of the latter the increased relative abundance of the bacterium in the probiotic-fed group can also be considered promoting intestinal health of the ducks. As for *Cellulosilyticum lentocellum*, another bacterium with increased abundance in the feces of the probiotic-fed ducks, it is known as a slow cellulose-degrader and member of the healthy animal fecal bacteriobiome [42]. Increased presence of an unclassified *Clostridiaceae_1* OTU, important anaerobic degraders of plant polymers [43], in the fecal bacteriobiome of probiotic-fed ducks, can be considered beneficial and hence might have contributed to higher production performance of the ducks. Overall, increased abundance of *Clostridiales* representatives in probiotic-fed ducks corroborates the use of these bacteria for novel probiotic formulations: recently some of the latter were shown to exert beneficial influence on Peking duck performance [44].

It should be noted that the available information about the influence of probiotic-enriched feed on ducks' gut microbiota is inconclusive, as both beneficial [28] and neutral effects were reported earlier [45]. One of the reasons for such a discrepancy may be because

two independent groups of birds, with and without probiotic supplementation, are usually compared: implementing the repeated measures design, i.e., sampling the same member of a group before and after the probiotic treatment, is more difficult in practice even in case of feces. Other reasons may be associated with regional/country differences between the studied duck groups such as lifespan, feed and its supplementation, medication, raising conditions, genetics [28,30], etc.

Compared with the α-biodiversity indices for the gut microbiota of ducks reported earlier [30], in our study Shannon and Chao-1 indices were lower. The dominance indices were found to show tendency to be lower in probiotic-fed group, while equitability tended to be higher. Therefore, overall α-diversity seemed to be increasing in the probiotic-fed group, which is generally regarded as positive.

Our finding that probiotic supplementation decreased feed consumption per unit of living mass of the ducks indirectly corroborates the results about increased abundance of beneficial, particularly plant material fermenting, bacteria, which most likely translated into more efficient transformation of nutrients in the gut and consequent more efficient utilization of nutrients by host organisms, i.e., ducks.

The found beneficial effect of probiotic-enriched feed on ducks' production characteristics agrees with the improved production performance of probiotic-fed Pitalah ducks [46]. The Pekin ducks' productivity performance in our study corroborates the beneficial influence of *Bacillus*-based supplementation on egg quality and biochemical properties of blood of Shaoxing ducks [47], and on gut microbiota established with lysine-yielding *Bacillus subtilis* on a locally domesticated Chinese duck breed [48]. Enhanced Pekin ducks' production, associated with beneficial changes in ducks' gut microbiota due to *Bacillus*-based probiotic supplementation, is also in line with improved growth performance shown by the Cherry Valley ducks [13]. Therefore, the studied *Bacillus*-based probiotic formulation a promising basis for further improvement [6] and use.

Finally, we want to stress that it is difficult to compare studies on duck intestinal micrtobiome diversity due to substantive differences in methodology, beginning from the studied groups (species, breed, age, raising conditions, site of sample collection in the gut, etc.) and all the way to amplification (primers), sequencing (platforms) and bioinformatic tools (software and databases). Therefore, there is an urgent need for a comprehensive meta-analysis of the duck gut microbiome data, hopefully resulting in recommendations for a more standardized research approach.

We also want to emphasize, albeit truistically, that case–control design, often used to infer the medication/supplementation-associated effects in humans and animals, prevents following directly, i.e., in one and the same individual, the dynamics of the properties of interest, i.e., bacteriobiome diversity as in our study. Therefore, repeated measures' design should be implemented if and when possible and feasible, despite the objective difficulties of doing so in studies with animals. Such a design helps to move closer to the cause–effect mechanisms of microbiome shifts, rather than be confined to association/correlation relations, as most of the microbiome studies do.

## 5. Conclusions

Our study aimed at comparing the structure and composition of fecal microbiota, as determined using 16S rRNA gene amplicon sequencing, in ducks receiving conventional and *Bacillus*-based probiotic supplemented feed. This is the first profile of gut bacteriobiome of domestic ducks in Russia and as such can be used as a regional reference in further research as well as a tiny contribution for constructing the global pattern. Duck fecal bacteriobiome was found to be drastically dominated by just two phyla (*Firmicutes* and *Proteobacteria*), represented by three classes (*Clostridia*, *Bacilli* and *Gammaproteobacteria*). *Escherichia/Shigella* sp. turned out to be the major and sole common dominant in all samples. Fecal bacteriobiome of probiotic-fed ducks differed from the conventionally fed control in the relative abundance of some dominant OTUs, mainly the pathogenic ones (*Escherichia/Shigella* sp., *Streptococcus* sp.). A number of minor and rare members of bacterial

assemblages (12% of the total number of OTUs) also displayed differential abundance; however, it was difficult to infer their physiological and/or pathogenic significance. The *Bacillus* bacteria, contained in the probiotic preparation used in the study, could not survive in the gut and were eliminated. Supplementation of the conventional feed with *Bacillus*-based probiotic resulted in pronounced shifts towards the more beneficial gut microbiota of ducks. The increased bacteriobiome α-diversity in the probiotic-fed group enhance gut microbiota and hence ducks' resilience towards adverse environmental effects. The bacterial OTUs, found to be the significantly related to the probiotic supplementation, provide a framework for further research on bacteria functioning and interactions within gut microbiota in order to improve birds' health and, as a consequence, both industrial and small farm poultry production. The studied *Bacillus*-based probiotic is promising for the development of improved formulations for specifically targeted interventions to modify gut microbiota of ducks. Such formulations can be effective alternatives for growth-promoting antibiotics, but there is still a great need to understand the role of poultry gut microbiota in the prophylaxis, growth and health promoting mechanisms.

**Author Contributions:** Conceptualization, V.I.P. and N.S.Z.; methodology, V.I.P.; software, M.R.K.; validation, T.Y.A., N.B.N. and A.V.K.; formal analysis, N.B.N.; investigation, A.V.K.; resources, V.I.P.; data curation, N.B.N.; writing—original draft preparation, N.B.N.; writing—review and editing, N.A.L., M.R.K. and V.I.P.; visualization, T.Y.A.; supervision, M.R.K.; project administration, V.I.P.; funding acquisition, V.I.P. All authors have read and agreed to the published version of the manuscript.

**Funding:** This research was funded by the MINISTRY OF SCIENCE AND HIGHER EDUCATION OF THE RUSSIAN FEDERATION (project number 121031300042-1).

**Institutional Review Board Statement:** The study was conducted according to the guidelines of the Declaration of Helsinki, and approved by the Institutional Review Board of STOLYPIN OMSK STATE AGRARIAN UNIVERSITY (protocol code 23/11 23 August 2018).

**Informed Consent Statement:** All experimental procedures involving ducks met the guidelines approved by the in-stitutional animal care and use committee and were performed in accordance with the Russian National Law concerning the care of animals for research purposes, as well as in compliance with the European Commission Directive 2010/63/EU on the protection of animals used for scientific purposes.

**Data Availability Statement:** The read data reported in this study were submitted to the GenBank under the study accession PRJNA523560.

**Acknowledgments:** 

**Conflicts of Interest:** The authors declare no conflict of interest. The funders had no role in the design of the study; in the collection, analyses, or interpretation of data; in the writing of the manuscript, or in the decision to publish the results.

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
