# Peer review of "Bacillus-Based Probiotic Treatment Modified Bacteriobiome Diversity in Duck Feces"

_agriculture, doi:10.3390/agriculture11050406_

Round 1

Reviewer 1 Report

The paper  is well written; the text is clear and easy to read; only minor revision is required.

Line 40 after chicken, mouse, and pig please add references. I suggest you  (Zamanizadeh, A.; Mirakzehi, M.T.; Agah, M.J.; Saleh, H.; Baranzehi, T. A comparison of two probiotics Aspergillus oryzae and, Saccharomyces cerevisiae on productive performance, egg quality, small intestinal morphology, and gene expression in laying Japanese quail. Ital J Anim Sci 2021, 20, 232-242 https://doi.org/10.1080/1828051X.2021.1878944;  Wang, H.; Ha, B.D.; Kim, I.H. Effects of probiotics complex supplementation in low nutrient density diet on growth performance, nutrient digestibility, faecal microbial, and faecal noxious gas emission in growing pigs. Ital J Anim Sci 2020, 20, 163-170 https://doi.org/10.1080/1828051X.2020.1801358;    

Liu, X.; Liu, W.; Deng, Y.;  He, C.; Xiao, B.; Guo, S.; Zhou, X.; Tang, S.; Qu, X.

Use of encapsulated Bacillus subtilis and essential oils to improve antioxidant and immune status of blood and production and hatching performance of laying hens   Use of encapsulated Bacillus subtilis and essential oils to improve antioxidant and immune status of blood and production and hatching performance of laying hens. Ital J Anim Sci 2020, 19, 1583-1591   https://doi.org/10.1080/1828051X.2020.1862715)

In materials and methods indicate the type of litter used

Table 4  delete “Feed intake, kg/bird”           unnecessary

Lines 209-210 Please delete the period  “Over 60 days of growth the ducks in the control group

consumed on average 0.72 kg more feed, as compared to the probiotic-fed ducks”

Author Response

Point 1

The paper  is well written; the text is clear and easy to read; only minor revision is required.

Response 1

Thank you very much!

Point 2

Line 40 after chicken, mouse, and pig please add references. I suggest…

Response 2

Thank you awfully for you suggestion; it was extremely helpful and relevant to the topic. The suggested references added in the text and reference list; the changes are highlighted in turquoise in the revised version.

Point 3

In materials and methods indicate the type of litter used

Response 3

Indicated as suggested (“pine-shavings deep litter”); the changes are highlighted in turquoise in the revised version.

Point 4

Table 4  delete “Feed intake, kg/bird”           unnecessary

Response 4

Deleted as suggested; the change is highlighted in turquoise in the revised version.

Point 5

Lines 209-210 Please delete the period  “Over 60 days of growth the ducks in the control group consumed on average 0.72 kg more feed, as compared to the probiotic-fed ducks”

Response 5

Deleted as suggested. The change is highlighted in turquoise in the revised version.

Reviewer 2 Report

In the manuscript the authors presented an important point, although the manuscript has some drawbacks.

Main remarks:

1. Introduction - very short. What is the research gap?

2. Literature review - after the "introduction", there should be a part of "literature review".

3. Conclusions - very short. In your conclusions, please also answer the following questions:

• what are the directions for the future?
• what are the research gaps?
• what is new to this manuscript?

Author Response

Point 1

Introduction - very short. What is the research gap?

Response 1

As it was stated in the section, there is little data on duck gut microbiome in commercially raised duck under Bacillus–based probiotic supplementation. Anyway, we added more references (some of them according to another reviewer’s suggestions), as well as  the text: “and we failed to find reports on gut microbiome research using probiotic Bacillus strains for Pekin ducks”. The changes are highlighted in turquoise in the revised version.

Point 2

Literature review - after the "introduction", there should be a part of "literature review".

Response 2

Strictly speaking, we saw nothing immediately relevant for reviewing and believed that asymptotic reviewing will make the introduction cumbersome and not to the point. However, we cited some studies in an attempt to make the current situation more explicit. To reinforce the aim, we added the text “Knowledge of the microbiome profiles in regional agricultural populations could help in drawing a global picture of duck gut microbiota, leading to a better insight of the regional effects of production technologies such as the use of probiotics, prebiotics, synbiotics, enzymes and antibiotics”. The changes are highlighted in turquoise in the revised version.

Point 3

Conclusions - very short. In your conclusions, please also answer the following questions:

• what are the directions for the future?

• what are the research gaps?

• what is new to this manuscript?

Response 3

As suggested, we added the sentences addressing the points. In conclusion we added “”This is the first profile of gut bacteriobiome of domestic ducks in Russia and as such can be used as a regional reference in further research as well as a tiny contribution for constructing the global pattern.” And at the very end of Conclusion: “Such formulations can be effective alternatives for growth-promoting antibiotics, but there is still a great need to understand the role of poultry gut microbiota in the prophylaxis, growth and health promoting mechanisms.”.

At the very end of Discussion section, we added the following text: “Finally, we want to stress that it is difficult to compare studies on duck intestinal micrtobiome diversity due to substantive differences in methodology, beginning from the studied groups (species, breed, age, raising conditions, site of sample collection in the gut etc.) and all the way to amplification (primers), sequencing (platforms) and bioinformatic tools (software and databases). Therefore there is an urgent need for a comprehensive meta-analysis of the duck gut microbiome data, hopefully resulting in recommendations for a more standardized research approach.

We also want to emphasize, albeit truistically, that case-control design, often used to infer the medication/supplementation-associated effects in humans and animals, pre-vents following directly, i.e. in one and the same individual, the dynamics of the properties of interest, i.e. bacteriobiome diversity as in our study. Therefore repeated measures’ design should be implemented if and when possible and feasible, despite the objective difficulties of doing so in studies with animals. Such design helps to get closer to the cause-effect mechanisms of microbiome shifts, rather than be confined to associa-tion/correlation relations, as most of the microbiome studies do”.  

The changes are highlighted in turquoise in the revised version.

Thank you very much for reviewing and for your suggestions to improve the logic and comprehensiveness of the manuscript!

Round 2

Reviewer 2 Report

Accept in present form. Good luck!